SciPost Physics

Submission

# Gate-induced decoupling of surface and bulk state properties in selectively-deposited Bi$_2$Te$_3$ nanoribbons

D. Rosenbach[1,2*†], K. Moors[1], A. R. Jalil[1] J. Kölzer[1,2] E. Zimmermann[1,2] J Schubert[1] S. Karimzadah[1] G. Mussler[1] P. Schüffelgen[1] D. Grützmacher[1,2] H. Lüth[1,2] Th. Schäper[1,2]

**1** Peter Grünberg Institute (PGI-9), Forschungszentrum Jülich, 52425 Jülich, Germany
**2** JARA-Fundamentals of Future Information Technology, Jülich-Aachen Research Alliance, Forschungszentrum Jülich and RWTH Aachen University, Germany
\* d.rosenbach@utwente.nl
†present address: MESA+ Institute for Nanotechnology, University of Twente, 7500AE Enschede, The Netherlands

August 3, 2021

## Abstract

Three-dimensional topological insulators (TIs) host helical Dirac surface states at the interface with a trivial insulator. In quasi-one-dimensional TI nanoribbon structures the wave function of surface charges extends phase-coherently along the perimeter of the nanoribbon, resulting in a quantization of transverse surface modes. Furthermore, as the inherent spin-momentum locking results in a Berry phase offset of $\pi$ of self-interfering charge carriers an energy gap within the surface state dispersion appears and all states become spin-degenerate. We investigate and compare the magnetic field dependent surface state dispersion in selectively deposited Bi$_2$Te$_3$ TI micro- and nanoribbon structures by analysing the gate voltage dependent magnetoconductance at cryogenic temperatures. Hall measurements on microribbon field effect devices show a high bulk charge carrier concentration and electrostatic simulations show an inhomogeneous gate potential profile on the perimeter of the TI ribbon. In nanoribbon devices we identify a magnetic field dependency of the surface state dispersion as it changes the occupation of transverse subbands close to the Fermi energy. We quantify the energetic spacing in between these subbands by measuring the conductance as a function of the applied gate potential and use an electrostatic model that treats the inhomogeneous gate profile and the initial charge carrier densities on the top and bottom surface. In the gate voltage dependent transconductance we find oscillations that change their relative phase by $\pi$ at half-integer values of the magnetic flux quantum applied coaxial to the nanoribbon providing evidence for a magnetic flux dependent topological phase transition in narrow, selectively deposited TI nanoribbon devices.

# 1   Introduction

Quasi-one-dimensional structures of three-dimensional topological insulators (TI) are of great interest, as they are predicted to host Majorana zero modes, when proximity coupled to an s-wave superconducting metal [1–3]. Two pairs of these exotic quasiparticle excitations can be used to encode the state of a topological quantum bit (qubit) [4, 5]. Large arrays of one-dimensional TI nanoribbons are envisioned for a scalable approach to define a quantum register of topological qubits [6, 7]. Novel epitaxial methods have been developed in order to grow TI nanoribbons selectively by molecular beam epitaxy (MBE) [8, 9]. On silicon hexagonal surfaces, partially covered by $SiO_2$ and amorphous $Si_3N_4$, this approach promises a high yield of selectively grown nanoribbons and highly scalable device networks.

The class of three-dimensional TI materials have Dirac surface states with linear dispersion and a unique helical spin texture as the spin of charge carriers is locked to their momentum [10–12]. When the phase-coherence length exceeds the perimeter of the nanoribbon, counterpropagating waves of surface charge carriers will self-interfere and their wave functions form standing waves that fit within the perimeter of the nanoribbon [13, 14]. As a result transverse-momentum subbands along the nanoribbon perimeter are quantized. Due to the inherent property of spin-momentum locking a Berry phase of $\pi$ is picked up by charges that perform one full rotation in momentum space (one full rotation along the nanoribbon perimeter) [10, 15]. The boundary conditions of self-interfering charge carriers are therefore antiperiodic and cause the surface state spectrum to be gapped in narrow nanoribbon structures [13]. In order to restore a pair of gapless, linear Dirac surface subbands a magnetic flux of $\Phi = (l + 1/2)\Phi_0$ needs to thread the cross section of the nanoribbon [16–18]. Here $\Phi_0 = h/e$ is the magnetic flux quantum and $l = 0, \pm 1, \pm 2, ..$ the transverse-mode index of quantized surface subbands. When the Dirac point resides close to the Fermi energy magnetic flux quantum-periodic Aharonov–Bohm-type (AB) magnetoconductance oscillations reflect the periodic appearance of the gapless, linear Dirac subbands as the system undergoes a topological phase transition [14, 19]. In bulk insulating TI nanoribbon devices, where only these linear Dirac subbands are populated by mobile charge carriers, a perfectly transmitted mode will establish as charge carriers can not scatter into states of opposite momentum and opposite spin [2, 3, 20].

Thin films and nanoribbons of TI materials often suffer from a high bulk charge carrier density [11, 21, 22]. AB-type magnetoconductance oscillations in bulk doped TI nanorib-

bons reflect the flux-periodicity of the surface state dispersion as the number of occupied transverse subbands below the Fermi energy changes [12, 16, 22–25]. The relative position of the quantized transverse subbands to the Fermi energy can be changed by accumulating or depleting charge carriers using a top gate-voltage, which effectively changes the occupation of transverse subbands along the nanoribbon perimeter $P$ at fixed magnetic fields. Gate voltage-dependent magnetoconductance oscillation patterns have previously been reported to quantify the energetic spacing of quantized transverse-momentum subbands in rectangular HgTe nanoribbons defined by wet-chemical etching [18].

In this research article we report on the electrical investigation of selectively-deposited $Bi_2Te_3$ micro- and nanoribbon field-effect devices at cryogenic temperatures. We first study the magnetic field and gate-voltage dependency of the selectively-deposited microribbon Hall bar devices. As the perimeter of the microribbon exceeds the phase-coherence length of surface charge carriers [22] no AB-type oscillations are observed in the magnetoconductance data of the wide ribbon device. We find that the gate tunability is partially obstructed by the high density of bulk charges as well as the asymmetric field-effect due to the device geometry. We proceed to investigate low-dimensional nanoribbon field-effect devices of selectively-deposited $Bi_2Te_3$. Despite the high bulk carrier density we identify surface-specific AB-type oscillations when applying a coaxial magnetic field. We analyse the coaxial magnetic field and gate voltage dependency of the transconductance within the nanoribbon devices and identify the flux dependent surface state dispersion as well as the transverse-momentum subband level spacing within the surface states of our selectively-deposited $Bi_2Te_3$ nanoribbon devices. Due to the inhomogeneous electric field distribution we consider an effective capacitance model [18] adapted to our highly bulk-doped nanoribbon devices. Our analysis shows evidence of quantized transverse-momentum states on the perimeter of the TI nanoribbon and the electrostatic model treatment allows to distinguish these features from bulk effects or conventional two dimensional space charge layers without spin-momentum locking.

## 2 Selectively deposited nanoribbon devices

### 2.1 Selective area epitaxy

Hall bar devices of different widths have been prepared by MBE following a selective area growth (SAG) approach [8, 9, 22]. Trenches of different widths $W$ have been defined in a layer stack of 20 nm $Si_3N_4$ and 5 nm $SiO_2$ layer on top of a Si(111) substrate. First the oxide layer is created by thermal oxidation of the surface of the Si(111) substrate. After the oxidation the 20 nm thick $Si_3N_4$ layer is deposited using a low pressure chemical vapor deposition process. The ratio of the layer thicknesses of the $SiO_2$ buffer layer and the $Si_3N_4$ layer is chosen in order to remove any strain these layers impose onto the Si(111) surface. The trenches are defined by a combination of wet- and dry-chemical etching using a positive electron beam resist. Reactive ion etching (RIE) (CHF$_3$ and O$_2$ gas mixture) has been used to transfer the trench structures from the electron beam resist into the nitride layer. After resist removal the $SiO_2$ within the nanotrenches is wet chemically removed using hydrofluoric (HF) acid, which renders the revealed Si(111) surfaces atomically smooth.

The $Bi_2Te_3$ binary TI has been grown in the Te-overpressure regime at $T_{sub}$=290 °C selectively within the defined nanotrenches. After deposition of the TI layer a 2-3 nm-thin $Al_2O_3$ layer is deposited using an electron beam evaporator and a stoichiometric target. The dielectric capping layer is used to protect the TI film from oxidation or other kinds of reactions with air.

## 2.2 Field effect in TI ribbon devices

A false-colored scanning electron micrograph of a $200\,\text{nm}$ wide ribbon Hall bar is shown in Fig. 1 a). An applied electric field (as schematically depicted in Fig. 1 b)) will simultaneously change the charge carrier density within the top surface (green), the bulk (red) and the bottom surface (green) of the $Bi_2Te_3$ field-effect device [26, 27]. An applied gate potential $V_g$ changes the two-dimensional charge-carrier density in a parallel plate capacitor geometry given by

$$\Delta n_{2D} = \frac{C}{e} \cdot \Delta V_g, \tag{1}$$

where $C$ is the capacitance of the field-effect device. The geometry of devices investigated induces a non-homogeneous electric field distribution and the capacitance on the nanoribbon perimeter $P$ becomes a position dependent value $C(s)$, where $s = [0, P]$. An effective capacitance model can be used to describe the gate effect on the surface state spectrum on the perimeter of a nanoribbon or nanowire geometry [18]. The average, gate dependent energy of the surface state charge carriers can be determined following

$$\langle E(s, V_g) \rangle = E_{DP} + \hbar v_F \sqrt{4\pi} \left\langle \sqrt{n_{2D}^{TSS}(s) + C(s)V_g/e} \right\rangle \tag{2}$$
$$= E_{DP} + \hbar v_F \sqrt{4\pi[n_{2D}^{TSS,av.} + C_{eff}^{TSS}V_g/e]},$$

where $v_F$ is the Fermi velocity, $\langle ... \rangle$ denotes the average along the nanoribbon perimeter $P$, $E_{DP}$ the Dirac point energy and $n_{2D}^{TSS}$ the charge carrier density on the topological surface states (TSSs). $n_{2D}^{TSS,av.}$ is the average charge carrier density on the topological surface states and $C_{eff}^{TSS}$ the effective capacitance. In systems where the surface state spectrum is initially pinned to the Dirac point ($n_{2D}^{TSS} = 0$) the equation simplifies to [18]

$$\langle E(s, V_g) \rangle = E_{DP} + \hbar v_F \sqrt{4\pi V_g C_{eff}/e}, \tag{3}$$

where the effective capacitance can be obtained by integration $C_{eff} = (1/P \int_0^P \sqrt{C(s)}ds)^2$. When the surface state spectrum is not initially pinned to the Dirac point, the initial average charge carrier density as well as the effective capacitance need to be considered. Molecular beam epitaxy grown $Bi_2Te_3$ micro- and nanoribbons [22] are unintentionally bulk doped during deposition. In such highly bulk conductive samples it can be assumed that the change in charge carrier density for small gate potentials applied is smaller than the initial charge carrier density $C(s)V_g \ll n_{2D}^{TSS}(s)$. With this assumption the effective capacitance in highly bulk doped systems can be calculated using

$$C_{eff} = \left\langle \sqrt{n_{2D}^{TSS}(s)} \right\rangle \left\langle C(s)/\sqrt{n_{2D}^{TSS}(s)} \right\rangle, \tag{4}$$

where the average charge carrier concentration can as well be determined through integration $\left\langle \sqrt{n_{2D}^{TSS}(s)} \right\rangle = n_{2D}^{TSS,av.} = (1/P \int_0^P \sqrt{n_{2D}^{TSS}(s)}ds)^2$.

## 2.3 Confinement in narrow TI nanoribbons

Here we investigate selectively deposited Hall bars of width $W = 1\,\mu\text{m}$ and $W = 200\,\text{nm}$. Due to the SAG approach the layer thicknesses vary slightly, dependent on the width of the nanoribbon [22], which measure $t = 10\,\text{nm}$ and $t = 15\,\text{nm}$ for the wide and narrow nanoribbon, respectively. The perimeter $P \approx 2\,\mu\text{m}$ of the wide nanoribbon is expected to be longer than the phase-coherence length of surface charges at $T = 1.5\,\text{K}$ [22]. The surface

states of the wide nanoribbon are therefore expected to resemble the Dirac dispersion as observed in slabs of $Bi_2Te_3$ (schematically depicted in Fig. 1 c), left) [11,28,29]. The phase-coherence length is however expected to be comparable to the perimeter $P = 430\,\text{nm}$ of the narrow nanoribbon. The confinement in the narrow nanoribbon device is expected to result in a quantization of transverse-momenta $k_l$ [13, 30, 31] (schematically depicted in Fig. 1 c), right). The coaxial- and transverse-momentum-dependent energy dispersion $E(k_x, k_l)$ within the confined nanoribbon structure is expressed by [13]

$$
\begin{aligned}
E(k_x, k_l) &= \pm \hbar v_\text{F} \sqrt{k_x^2 + k_l^2} \\
&= \pm \hbar v_\text{F} \sqrt{k_x^2 + \left(\frac{2\pi(l + 1/2 - \Phi/\Phi_0)}{P}\right)^2},
\end{aligned}
\tag{5}
$$

where $k_x$ is the coaxial-momentum and $k_l = 2\pi(l+1/2-\Phi/\Phi_0)/P$ the confined transverse-momentum, with $\Phi_0 = h/e$ being the magnetic flux quantum. The nanoribbon dispersion is characterised by quantized transverse-momentum subbands of quantum number $l = 0, \pm 1, \pm 2, \dots$ . Only for an applied magnetic flux $\Phi/\Phi_0 = l + 1/2$ there is a state at zero

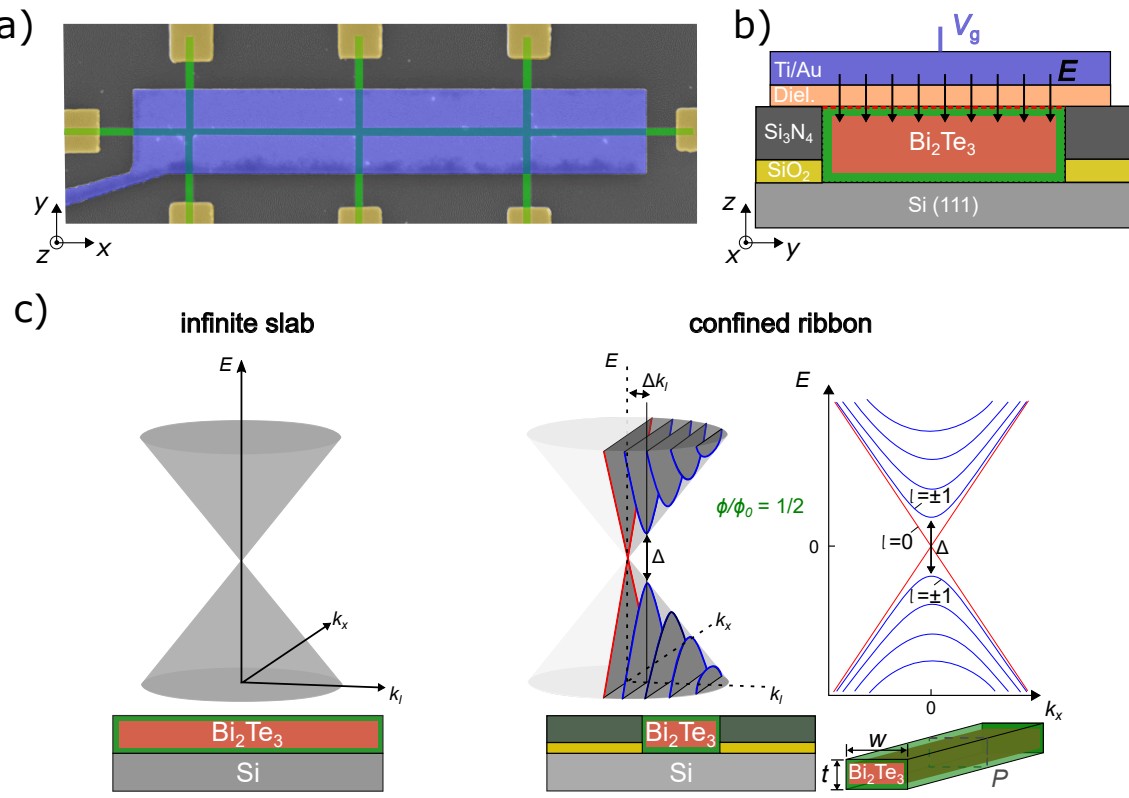

Figure 1: Dispersion relation in thin films and in confined nanoribbons of $Bi_2Te_3$. a) A false-colored scanning-electron micrograph of a gated, 200 nm-wide Hall bar is shown. The contact electrodes on the TI (green) are highlighted in ochre and the top gate electrode is highlighted (blue). b) The device geometry including the SAG mask (yellow and dark grey), the $Bi_2Te_3$ nanoribbon (green and red highlighting the surface and bulk regions, respectively), the gate dielectric (rose) and the top gate electrode (blue) are schematically shown. c) The dispersion $E(k_x, k_l)$ of the surface states on the surfaces of an infinite slab of $Bi_2Te_3$ represent a Dirac cone. In narrow nanoribbon geometries the subband level spacing $\Delta k_l = 2\pi/P$ increases with decreasing nanoribbon perimeter $P = 2 \cdot (W + t)$, where $W$ and $t$ are the width and the thickness of the nanoribbon, respectively.

energy. Without applied magnetic flux or at other values of the magnetic flux applied the surface state dispersion features a finite energy gap around zero. The size of this energy gap in the surface state spectrum of quantized transverse modes is given by [13]

$$\Delta = \frac{2\pi v_F \hbar}{P}. \tag{6}$$

For an applied magnetic flux of $\Phi = \Phi_0/2 = h/2e$ the energy dispersion for the $l = 0$ transverse-momentum state is linear and offers zero-energy solutions. A pair of gapless, linear Dirac surface subbands establishes. When the magnetic flux further increases the surface state spectrum will again be gapped. The topological phase transition can be observed with a period of one full integer flux quantum [14].

# 3 Electrical characterization of micro- and nanoribbon field effect devices

## 3.1 Gate-dependent microribbon Hall measurements

Devices have been characterized usig a variable temperature insert (VTI) cryostat with $1.5\,\mathrm{K}$ base temperature. Magnetic fields of up to $13\,\mathrm{T}$ field strength can be applied perpendicular and coaxial to the TI ribbon. By applying an a.c. current bias along the microribbon Hall bar the longitudinal magnetoresistance $R_{xx}$ as well as the Hall resistance $R_{xy}$ have been determined in a perpendicular magnetic field using standard lock-in techniques. The Hall resistance $R_{xy}$ and the longitudinal magnetoresistance $R_{xx}$ have been determined as a function of the gate voltage $V_g$. For the $1\,\mu$m-wide ribbon Hall bar a $15\,\mathrm{nm}$-thick $HfO_2$ ($\epsilon_r = 18.75$) dielectric layer has been deposited by atomic layer deposition. Using a source-meter the leakage current has been determined to be negligible up to a top gate voltage of $|V_g| \le 16\,\mathrm{V}$ ($I_{\mathrm{leakage}} \le 1\,\mathrm{nA}$).

The anticipated initial band alignment is schematically depicted in Fig. 2 a). The bulk band gap for $Bi_2Te_3$ measures about $E_{\mathrm{gap}} = 165\,\mathrm{meV}$ [28]. In previous measurements on selectively-deposited $Bi_2Te_3$ Hall bar structures it has been observed that the bands on the bottom surface bend towards the $p$-type Si(111) substrate [22] and the Fermi energy from an analysis of Shubnikov–de Haas oscillations has been determined to reside $90\,\mathrm{meV}$ above the Dirac point, which is buried within the bulk valence bands [32]. On the top surface angle-resolved photoemission spectra (ARPES) show that the Fermi energy resides within the bulk conduction band, about $70\,\mathrm{meV}$ above the bulk conduction band minimum [32,33]. By performing Hall measurements the initial charge carrier density in the investigated devices can be determined. From Hall measurements at $V_g = 0$ (shown in Fig. 2 b), grey curve) the Hall slope is determined ($A_H = dR_{xy}/dB$) and the two-dimensional charge carrier density has been calculated to be $n_{2D} = (A_H e)^{-1} = 8.5 \times 10^{13}\,\mathrm{cm}^{-2}$. In Hall measurements the combined charge carrier density of the materials bulk and the surface states is probed. From previous studies of Shubnikov–de Haas oscillations the charge carrier density on the bottom surface has been identified as $n_{2D,\mathrm{bot}} = 5.3 \times 10^{11}\,\mathrm{cm}^{-2}$ [22]. From ARPES measurements a top surface charge carrier density of $n_{2D,\mathrm{top}} = 8.2 \times 10^{12}\,\mathrm{cm}^{-2}$ can be inferred [32,33] leaving a bulk charge carrier density of $n_{2D,\mathrm{bulk}} = 7.6 \times 10^{13}\,\mathrm{cm}^{-2}$. Results therefore indicate a high density of bulk charges. The intrinsic $n$-type doping of the bulk is due to Te antisite defects in $Bi_2Te_3$ [11,21]. Assuming an effective mass of $m^* = 0.58\,m_e$ [34] the Fermi energy in the bulk can be estimated to lie about $100\,\mathrm{meV}$ above the conduction band minimum.

When a negative gate voltage $V_g < 0$ is applied to the top gate electrode, electronic charges on the top surface of the TI will be depleted. As a result, the Hall slope in Fig. 2 b) increases. Simultaneously, the longitudinal resistance $R_{xx}$, shown in the inset of Fig. 2 b), increases as well. Charge carrier density as well as mobility values ($\mu = L \cdot (W R_{xx} A_H e)^{-1}$ with $L$ being the channel length), as determined from the $R_{xx}(V_g)$ and $R_{xy}(V_g)$ data, are shown as a function of the applied gate voltage in Fig. 2 c). In between $0\,\mathrm{V} \geq V_g \geq -5\,\mathrm{V}$ the charge-carrier density decreases linearly. As the amount of charge-carriers decreases, the mobility value increases gradually from an initial value of about $300\,\mathrm{cm}^2/\mathrm{Vs}$ at zero gate voltage to a value of about $360\,\mathrm{cm}^2/\mathrm{Vs}$ at $V_g = -5\,\mathrm{V}$. The increase in mobility can be explained as bulk scattering of surface charges on the top surface is reduced [35]. From a linear fit to the gate-dependent charge-carrier density (following Eq. 1) in between $-5\,\mathrm{V} \geq V_g \geq 0\,\mathrm{V}$ (dashed black line), a capacitance of $C_{\mathrm{exp}} = 9.8 \times 10^{-3}\,\mathrm{Fm}^{-2}$ can be esti-

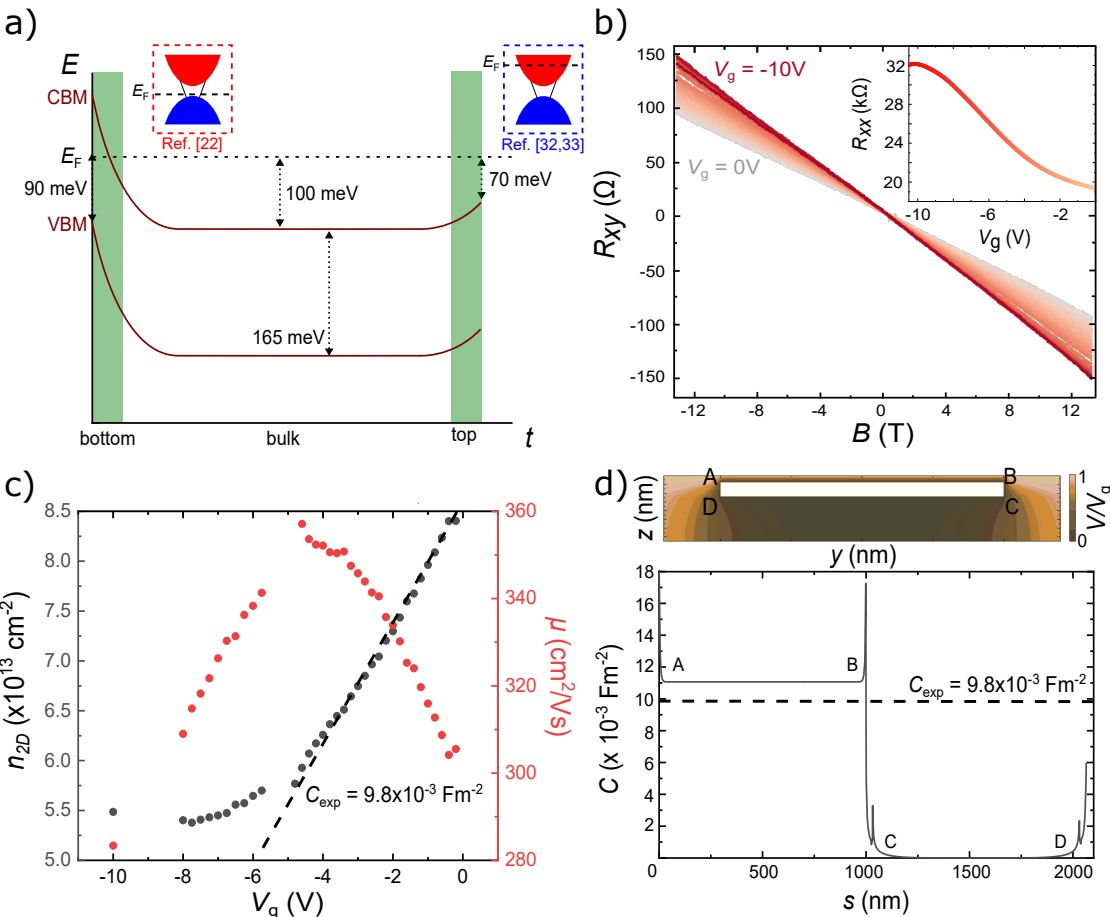

Figure 2: Gate-dependent Hall- and longitudinal resistance of the $1\,\mu$m wide Hall bar. a) Expected relative position of the Fermi energy within the bulk as well as band bending on top and bottom surface of the ribbon device. b) Hall resistance $R_{xy}$ measurements at different gate voltages $V_g$. The extracted Hall slopes $dR_{xy}/dB$ have been used to evaluate the sheet carrier concentration $n_{2D}$ shown in c) (black dots). The mobility $\mu$ values (red dots) have been evaluated by considering the gate-dependent sheet resistance $R_S$, extracted from the gate-dependent longitudinal resistance $R_{xx}(V_g)$ at zero magnetic field shown in the inset of b). In d) the simulated relative gate potential (top) and calculated capacitance (bottom) along the microribbon perimeter $P$ are displayed. The dashed black line represents the experimentally determined capacitance from c).

mated. Given this experimentally-determined capacitance the charge-carrier density on the top surface will be depleted at $\Delta V_{\mathrm{g}} = (e \cdot 8.2 \times 10^{16} \, \mathrm{m}^{-2})/9.8 \times 10^{-3} \, \mathrm{Fm}^{-2} = 1.3 \, \mathrm{V}$. As the linear trend of the gate voltage dependent charge-carrier density exceeds this value it can be concluded that both bulk and surface charges are depleted simultaneously. In the regime below $V_{\mathrm{g}} \leq -5 \, \mathrm{V}$ the charge-carrier density saturates, while the mobility values decrease gradually. It can be assumed that the Fermi energy on the top surface drops below the conduction band, rendering charge accumulation more difficult [27, 36]. Since the Hall slope does not change sign the majority of bulk carriers, however, remains $n$-type throughout the range of gate voltages applied.

The experimentally-determined capacitance of the microribbon field-effect device is compared to the geometrical capacitance of the device. The relative gate potential $V/V_{\mathrm{g}}$ over the perimeter of the device as shown in Fig. 1 c) is simulated. A generalized Poisson solver is applied to a finite-element mesh of 35.000 points on a regular square lattice of 1 nm spacing in the $yz$-plane (for details on the simulation of the electrostatic behavior of the devices, the reader is referred to Appendix A). For the simulation the surface of the material is treated as a perfect metallic conductor i.e., the electric field cannot penetrate the interior of the microribbon. The results are shown in the top part of Fig. 2 d). A line cut (A-B-C-D-A) along the microribbon perimeter $P$ is extracted and the local capacitances $C(s)$ are calculated. The results are shown in the bottom part of Fig. 2 d) including the experimentally determined capacitance $C$ (dashed black line). This value is compared to the effective capacitance calculated from the simulated capacitance profile on the perimeter of the nanoribbon using Eq. 4. The calculated effective capacitance $C_{\mathrm{eff}} = 3.6 \times 10^{-3} \, \mathrm{Fm}^{-2}$ is smaller than the experimentally determined capacitance $C_{\mathrm{exp}} \gg C_{\mathrm{eff}}$ due to the high density of bulk charges and the effective screening of the electric field on the bottom surface. As only charge carriers close to the surface of the microribbon are depleted the system resembles a parallel plate capacitor geometry. This includes charge carriers on the top surface as well as bulk charge carriers that reside close to the surface. The difference in the experimentally determined capacitance and the calculated capacitance on the top surface of the microribbon can be explained by small differences of the dielectric constants or the layer thicknesses of the layers involved compared to the values used in the simulation.

## 3.2 Magnetic field and gate voltage dependent conductance oscillations in nanoribbons

For the narrow nanoribbon Hall bar a 100 nm-thick LaLuO$_3$ ($\epsilon_r = 32$ [37]) dielectric layer has been deposited by pulsed-laser deposition. Applied gate voltages result in a smaller change of the charge carrier density compared to previously analysed microribbon field effect devices. This results in a better resolution when investigating electric and magnetic field dependent transconductance oscillations. Leakage currents in the nanoribbon field-effect devices are negligible up to gate voltages of $|V_{\mathrm{g}}| \leq 35 \, \mathrm{V}$.

Due to confinement, the topological surface-state spectrum is expected to be quantized, forming transverse-momentum subbands [13] following Eqs. 5 and 6. The magnetic flux dependency of the quantized subband dispersion of topological surface charges along the perimeter of the nanoribbon result in Aharonov–Bohm (AB) [38, 39] type oscillations that have previously been reported for different topological insulator materials [16, 23, 25, 40–42] and also recently for selectively-deposited Bi$_2$Te$_3$ nanoribbon devices of different cross sectional areas [22]. For a 200 nm wide Bi$_2$Te$_3$ nanoribbon field-effect device presented here the longitudinal magnetoconductance $G_{xx}(B)$ (black curve) has been measured in magnetic fields up to 13 T and equivalently show AB-type oscillations, as exemplary shown in Fig. 3 a) at $V_{\mathrm{g}} = -10V$. These oscillations have been measured to be reproducible after

two alternate cooldown cycles and can be distinguished from universal conductance fluctuations by studying the dependency of the oscillation frequency with respect to the angle span in between the nanoribbon and the applied magnetic field [22]. Due to a strong background, mainly due to the weak antilocalization (WAL) effect, the AB-type oscillations are better visible, when subtracting a slowly varying background ($\delta G_{xx}(B)$, red curve). The background is created using a Savitzky–Golay filter with an averaging window of 3001 data points (corresponding to a range of $3\,\mathrm{T}$). The applied magnetic flux $\Phi = B \cdot S$, where $S$ is the cross section of the nanoribbon, is displayed normalized to the magnetic flux quantum $\Phi_0$. As observed before [22], the cross sectional area determined from the flux periodicity $S = 2 \times 10^{-15}\,\mathrm{m}^2$ of the AB-type oscillations is slightly smaller than the geometrically determined cross sectional area $S = 2.6 \times 10^{-15}\mathrm{m}^2$. This has previously been attributed to the actual penetration depth of the wave function of the topological surface states [43], effectively reducing the cross sectional area. When applying a gate voltage, the position of the quantized transverse-momentum subbands with respect to the Fermi energy changes. The transconductance through the nanoribbon changes periodically dependent on the relative position of the subbands. Minima in the gate voltage dependent transconductance

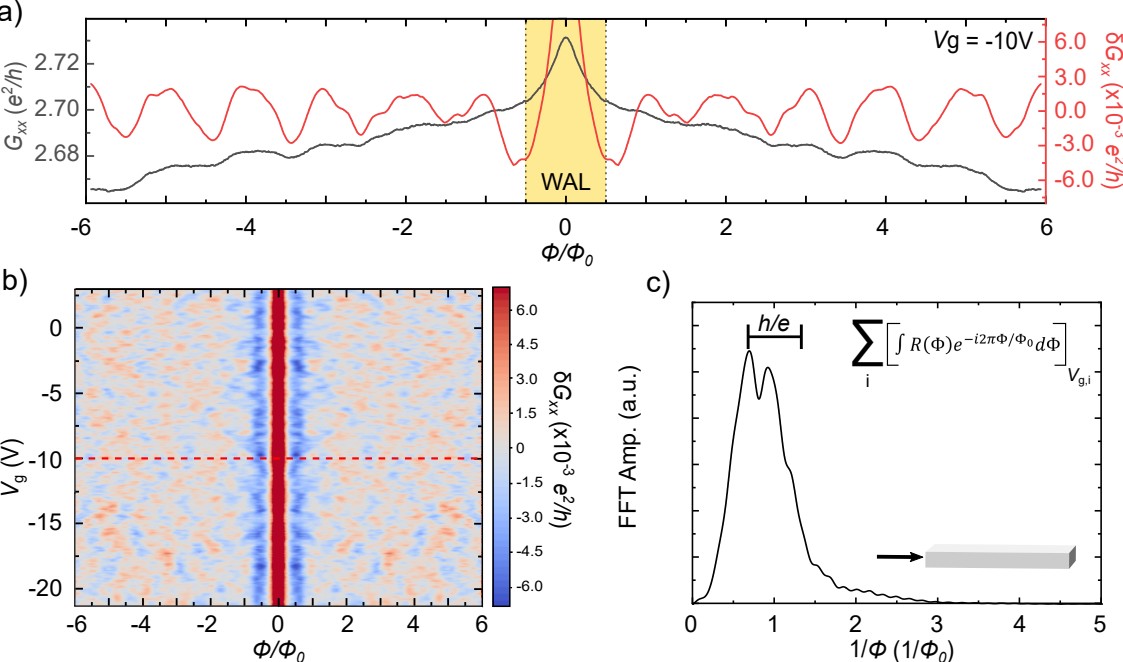

Figure 3: Gate-dependent magnetoconductance oscillations of the narrow nanoribbon in a parallel applied magnetic field. a) The magnetoconductance of a narrow nanoribbon device shows periodic Aharonov–Bohm oscillations, exemplary shown at a gate voltage of $V_\mathrm{g} = -10\,\mathrm{V}$ before (black curve) and after subtracting a slowly varying background (red curve). The low field is governed by the weak antilocalization effect (WAL). b) A set of magnetoconductance measurements performed at different gate voltages $V_\mathrm{g}$. Each $\delta G_{xx}$ curve shows a dominant WAL feature at low fields ($\Phi/\Phi_0 \sim 0$, red area) as well as flux quantum-periodic, symmetric oscillations at higher magnetic field strengths ($|\Phi/\Phi_0| > 0$). The flux-quantum periodicity of observed oscillations is further verified in the fast Fourier transformation performed on each dataset. In c) the sum of all FFT amplitudes of each respective scan are shown. The scale bar highlights the error of expected frequencies for the observation of flux-quantum periodic AB-type oscillations due to mentioned geometric uncertainty.

$G(V_{\mathrm{g}})$ thereby correspond to the Fermi energy residing at the edge (energetic minimum) of a transverse-momentum subband. At these points scattering is enhanced due to an increase in the density of states (van Hove singularities) [18]. When half a flux quantum is applied the energetic position of the subband minima are maximally shifted and the transconductance oscillations as a function of the applied gate voltage are shifted by $\Delta\phi = \pi$ [13,19]. In the following the dependency of the magnetoconductance as a function of an applied gate voltage and an applied coaxial magnetic field $G(V_{\mathrm{g}}, B)$ in the narrow nanoribbon device are investigated. Following the geometry argument (cf. Eq. 6) the anticipated spacing of individual subbands measures $\Delta = 4.5\,\mathrm{meV}$.

For gate voltages in between $-21\,\mathrm{V} \geq V_{\mathrm{g}} \geq 3\,\mathrm{V}$ the magnetoconductance modulations (after subtracting a smooth background) as a function of the applied magnetic flux and the gate voltage $\delta G_{xx}(\Phi, V_{\mathrm{g}})$ are shown in Fig. 3 b). The low-field data in between $-0.5 \geq \Phi/\Phi_0 \geq 0.5$ is dominated by the WAL effect and is generally neglected in the following analysis. At higher magnetic fields AB-type oscillations that vary with the applied gate voltage can be observed. For each magnetic field sweep a fast Fourier transformation (FFT) is performed and the sum of all resulting FFT amplitudes is taken. The results are shown in Fig. 3 c) and show a clear peak around $1/\Phi = 1/\Phi_0$ corresponding to AB-type oscillations of period $\Phi_0$. The error bar in the graph takes into account the deviation of the geometrically defined cross sectional area. The amplitude of these oscillations in quasi-ballistic systems is anticipated to be $e^2/h$ [13,24,43] as each subband acts as a single ballistic channel. In measurements presented here the observed AB-type oscillations have an amplitude of a fraction of a single conductance quantum. Reason therefore is that the channel length $L = 5\,\mu\mathrm{m}$ is much longer than the elastic mean free path of surface charges [22].

Observed $\Phi_0$-periodic AB oscillations show that the surface charges interfere on the nanoribbon perimeter phase-coherently. In order to quantify the phase-coherence length of surface state charges on the nanoribbon perimeter we evaluate the temperature dependency of the AB-type oscillations at a fixed gate voltage of $V_{\mathrm{g}} = -12\,\mathrm{V}$. The magnetoconductance modulations $\delta G_{xx}$ in between $T = 1.5\,\mathrm{K}$ base temperature up to $T = 30\,\mathrm{K}$ are shown in Fig. 4 a). The AB oscillation amplitudes for two different peak positions (black square and red circle) are extracted and plotted as a function of the temperature $T$ in Fig. 4 b). From the decay of the AB oscillation amplitude the phase-coherence length can be estimated as $\delta G(T) \propto \exp(-P/l_\phi(T))$ [44]. The $\delta G_{xx}(T)$ data has been fitted using $\delta G = \delta G_0 \exp(-aT^{1/2})$, as $l_\phi \propto T^{-1/2}$ [45]. The phase-coherence length at $T = 1\,\mathrm{K}$ can be estimated to be $l_\phi = (360 \pm 30)\,\mathrm{nm}$.

After identifying phase-coherent AB oscillations on the nanoribbon perimeter we now focus on the quantitative description of the subband level spacing of confined states. Analogous to the analysis procedure described by Ziegler et al. [18] we take the average of line cuts taken at multiples of integer values $i \cdot \Phi_0$ and multiples of half-integer values $(i + 1/2) \cdot \Phi_0$ of the magnetic flux quantum from Fig. 3 a). In this analysis the zero flux and the $\pm\Phi_0/2$ line cuts are excluded, due to the strong influence of the WAL feature in this region. The resulting curves are shown in Fig. 5 a) for applied gate voltages in the range of $-2\,\mathrm{V} \geq V_g \geq -12\,\mathrm{V}$. Both curves show a clear anticorrelated behavior, where maxima in the red curve coincide at the same gate voltage with minima of the black curve. As discussed before these minima in the transconductance as a function of the gate voltage are due to van Hove singularities at the edge of each transverse-momentum subband [18]. The anticorrelated maxima (minima) in the red (black) curve are indexed as indicated in

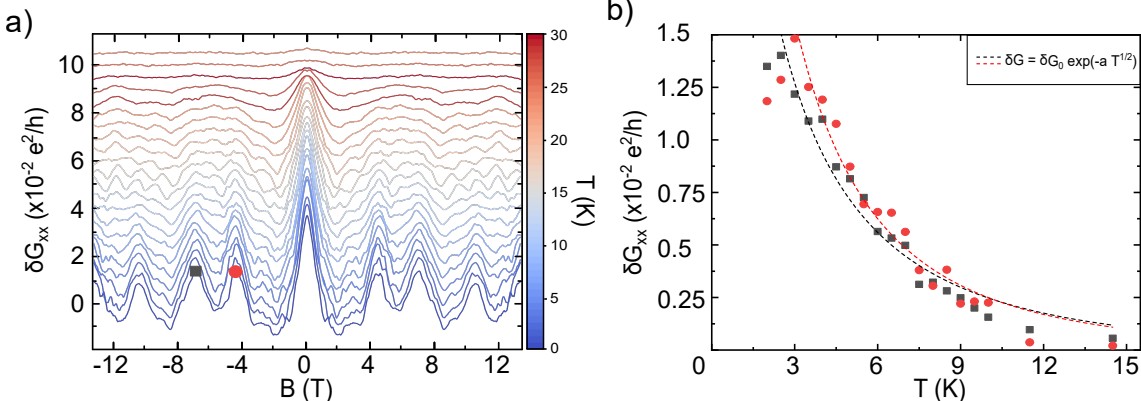

Figure 4: Temperature dependency of AB-type oscillations at $V_{\mathrm{g}} = -12\,\mathrm{V}$. a) The background subtracted magnetoconductance data $\delta G_{xx}$ of the narrow gated nanoribbon is shown at different temperatures. The temperature-dependent oscillation amplitude at $B = \pm 4.15\,\mathrm{T}$ (black circles) and at $B = \pm 7.1\,\mathrm{T}$ (red circles) is shown in b). An exponential fit has been performed to both datasets. An exponential fit as described in the main text is performed on both curves and the best fits are highlighted as a dashed black and dashed red line, respectively.

Fig. 5 a). The running index $N$ is thereby a relative value with respect to the number of occupied subbands $N_0$ at $V_{\mathrm{g}} = 0$. Since for Bi$_2$Te$_3$ the Dirac point lies within the bulk valence band ($E_{\mathrm{F}} - E_{\mathrm{DP}} \approx 300\,\mathrm{meV}$) [12, 28, 32, 33] it is not possible to identify the initial number of occupied subbands. The number of occupied subbands is, however, effectively reduced by applying a negative gate voltage $V_{\mathrm{g}} < 0$. In the range of applied gate voltages ($0\,\mathrm{V} \geq V_g \geq -21\,\mathrm{V}$) a total of more than 60 anticorrelated extrema have been identified, where the energetic spacing in between two subbands has been estimated to measure about $\Delta = 4.5\,\mathrm{meV}$ (cf. Eq. 6). It is therefore likely that within the range of gate voltages applied ($N \cdot \Delta = 270\,\mathrm{meV}$) the Dirac point moves close to the average Fermi energy. The indexed anticorrelated minima are displayed versus the absolute value of the gate voltage applied in Fig. 5 b). At every conductance minimum in the gate-dependent transconductance the Fermi wavevector can be related to the amount of occupied subbands by $k_F = k_0 - N\Delta k_l$. Depending on the spin-degeneracy of the system, the Fermi wave vector can also be expressed following $k_F = \sqrt{(4\pi/g_s)n_{2\mathrm{D}}^{\mathrm{TSS}}}$, with the spin degeneracy factor $g_{\mathrm{s}} = 1$ for topologically non-trivial and $g_{\mathrm{s}} = 2$ for topologically trivial states. The gate-dependent subband filling can therefore be expressed following [18]

$$V_{\mathrm{g}} - V_0 = \frac{g_s e}{4\pi C_{\mathrm{eff}}^{\mathrm{TSS}}} \left[ 2k_0(N - N_0)\Delta k_l + (N - N_0)^2 \Delta k_l^2 \right]. \tag{7}$$

The initial values are taken $V_0 = 0$ and $N = N_0$ to consider the relative change from the initial occupation of subbands at zero gate voltage. The initial Fermi wave vector can be determined from the effective charge-carrier density $k_0 = \sqrt{4\pi n_{2\mathrm{D}}^{\mathrm{TSS,av.}}(V_g = 0)} = 0.09\,\text{Å}^{-1}$. The subband spacing within the model in the above equation can be approximated from the effective geometry of the nanoribbon as obtained from the periodicity of the AB oscillations $\Delta k_l = 2\pi/P = 1.55 \times 10^7\,\mathrm{m}^{-1}$. The only free parameter left for the fit is the effective capacitance $C_{\mathrm{eff}}$. For $g_{\mathrm{s}} = 1$, indicating the subband spacing in topologically non-trivial surface states, the best fit performed is shown in Fig. 5 b), blue curve. The effective capacitance from the fit measures $C_{\mathrm{eff}}^{\mathrm{TSS}} = 1.4 \times 10^{-3}\,\mathrm{Fm}^{-2}$. For absolute gate voltages above $|V_{\mathrm{g}}| \geq 12\,\mathrm{V}$ the gate voltage needed to shift another subband through the

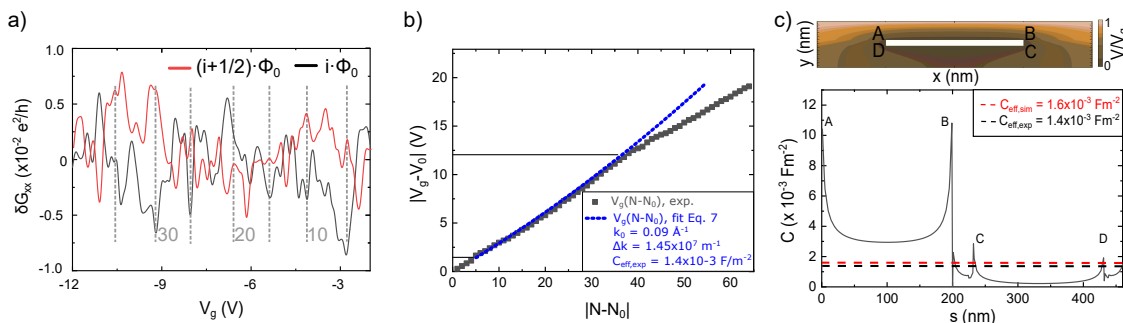

Figure 5: Gate-dependent subband spacing in the narrow nanoribbon. a) The average oscillation pattern of line-cuts at full integer (black curve) and half-integer (red curve) values of the magnetic flux quantum extracted from Fig. 4 a) are shown. The grey dashed lines show the extracted subband indices $N$, which are shown in b) as a function of the absolute applied gate voltage $V_g$. The black lines indicate the range of values shown in a). The blue dashed line indicates the fit performed following Eq. 7 and the values for the best fit performed are mentioned within the inset. c) Simulated relative gate potential $V/V_g$ along the perimeter of the nanoribbon (top) and calculated local capacitances $C(s)$ (bottom).

Fermi level decreases. When the Dirac point moves closer to the Fermi energy the charge carrier density within the surface states changes more abruptly and the approximation to get Eq. 4 is no longer valid. Therefore the evaluation of the effective capacitance is restricted to the range of gate voltages in between $3\,\text{V} \leq |V_g| \leq 12\,\text{V}$.

The experimentally determined value for the effective capacitance is compared to the effective capacity determined from the electrostatic model discussed in Eqs. 1-4. The relative gate potential on the nanoribbon perimeter is determined using previously mentioned Poisson solver. The results of the simulations are shown in Fig. 5c) (top) and include the calculated values for the local capacitances $C(s)$ (bottom). From these values the effective capacitance has been evaluated to be $C_{\text{eff,sim}} = 1.6 \times 10^{-3}\,\text{Fm}^{-2}$, which matches the experimentally determined effective capacitance quite well. Within the graph both the experimentally determined effective capacitance $C_{\text{eff,exp}}$ (black dashed line) and the effective capacitance determined from our electrostatic model $C_{\text{eff,sim}}$ (red dashed line) are highlighted.

## 4   Conclusions

We have electrically characterized selectively-deposited $Bi_2Te_3$ micro- and nanoribbon field-effect devices at cryogenic temperatures and used an electrostatic model to investigate the geometry dependence of the topological surface state dispersion. Our model considers that in Hall measurements on the microribbon Hall bar device, the $Bi_2Te_3$ layer has been identified to be strongly bulk conductive, which affects the gate tunability of topological surface charges on the perimeter of the device. Additionally, the device geometry results in an inhomogeneous gate potential profile, with the bottom surface being effectively screened from the electric field of the top gate potential. For that reason the change in charge carrier density of the microribbon Hall bar device is captured by a parallel plate capacitor model. Surface charges and bulk charges close to the top surface are being depleted simultaneously, which is verified by comparing the effective capacitance determined from the gate voltage

dependent change of the charge-carrier density to the calculated capacitance on the top surface. This model is limited by the observation that the charge carrier density decreases linearly first but then saturates. The saturation most likely occurs as charges from the top surface are being depleted while the bulk of the device remains highly conductive. The majority of charge-carriers in Hall measurements stay $n$-type throughout the whole range of gate voltages applied.

Unlike the wide microribbon field-effect device phase-coherent states span around the perimeter of the nanoribbon. Due to the geometrically well defined coherent conductance paths on the perimeter of the nanoribbon these transverse-momentum states are flux sensitive, resulting in magnetic flux quantum-periodic AB-type oscillations. The phase-coherence length from the temperature dependency of the AB oscillation amplitude $l_\phi = 360 \pm 30$ nm has been determined to be comparable to the perimeter $P$ of the nanoribbon device only. We identify the energetic spacing of the quantized transverse-momentum subbands, which corresponds well with the energetic spacing determined from geometric considerations [14]. We evaluate the magnetic flux dependent surface state dispersion and identify clear anti-correlated conductance maxima and minima in the gate voltage dependent transconductance at full-integer $\Phi = i \cdot \Phi_0$ and half-integer $\Phi = (i + 1/2) \cdot \Phi_0$ values of the magnetic flux quantum applied coaxial to the nanoribbon. In the analysis of the gate voltage dependent occupation of these quantized states we use a spin degeneracy factor of $g_s = 1$ [18]. We find that the effective capacitance from this analysis compares well with the effective capacitance determined using our electrostatics model. Results provide evidence for a magnetic-flux dependent topological phase transition in our narrow TI nanoribbons, which provides an odd number of surface-state band crossings at half-integer values of the magnetic flux quantum. This is an important requirement for the realization of gate-tunable Majorana devices that are partially covered by a superconducting metal [2]. The asymmetric chemical potential along the nanoribbon perimeter within such devices has just recently been reported to be advantegeous for the detection of Majorana bound states [46]. We show that our selectively deposited TI nanoribbon devices can be used to fabricate highly-scalable and gate-tunable networks of quasi-1D TI nanoribbon structures and networks for Majorana experiments [6,7,47], despite the presence of a high bulk background doping typically identified in these molecular beam epitaxy grown devices.

# Acknowledgements

This work was partly funded by the Deutsche Forschungsgemeinschaft (DFG, German Research Foundation) under Germany's Excellence Strategy - Cluster of Excellence Matter and Light for Quantum Computing (ML4Q) EXC 2004/1 - 390534769. This work was financially supported by the German Federal Ministry of Education and Research (BMBF) via the Quantum Futur project "MajoranaChips" (Grant No. 13N15264) within the funding program Photonic Research Germany.

**Author contributions**   D.R., J.K., E.Z. and A.R.J. have fabricated substrates for the selective area epitaxy in the cleanroom. A.R.J, G.M. and P.S. have performed the selective area epitaxy of the topological insulator thin film using molecular beam epitaxy. J.S. and S.K. have deposited the $LaLuO_3$ and $HfO_2$ dielectric layer using pulsed laser deposition and atomic layer deposition, respectively. D.R. has performed the electrical characterization of

devices using a variable temperature insert cryostat. K.M. has performed the simulation of the relative gate potential and the modelling of the effective capacitance. The project has been supervised by and discussed with D.G., H.L., and Th.S.

**Funding information**   This work was financially supported by the Virtual Institute for Topological Insulators (VITI), which is funded by the Helmholtz Association (VH-VI-511). This work was financially supported by the German Federal Ministry of Education and Research (BMBF) via the Quantum Futur project "MajoranaChips" (Grant No. 13N15264) within the funding program Photonic Research Germany. This work was partly funded by the Deutsche Forschungsgemeinschaft (DFG, German Research Foundation) under Germany's Excellence Strategy—Cluster of Excellence Matter and Light for Quantum Computing (ML4Q) EXC 2004/1—390534769.

## Appendix A:

To resolve the shift of the charge density locally on the nanoribbon perimeter, we solve the generalized Poisson equation for an inhomogeneous dielectric medium:

$$\nabla \cdot [\epsilon(\mathbf{r})\nabla V(\mathbf{r})] = 0 \tag{8}$$

The boundary conditions are given by $V(\mathbf{r}) = V_{\mathrm{g}}$ at the surface of the metal gate, and $V(\mathbf{r}) = 0$ at the the nanowire surface (assumed to be a perfect metal that completely screens the electric field in the interior with a shift of charge density on the surface). For the remaining (artificial) boundaries of the simulated region, we consider Neumann boundary conditions, $\mathbf{n} \cdot \nabla V(\mathbf{r}) = 0$ (corresponding to dielectric with $\epsilon \to +\infty$), with $\mathbf{n}$ the unit vector normal to the boundary, keeping the value of the potential floating. At the interface between two insulating materials with different dielectric constants, the following relation holds:

$$\epsilon_1 \mathbf{n} \cdot \nabla V(\mathbf{r})|_1 = \epsilon_2 \mathbf{n} \cdot \nabla V(\mathbf{r})|_2, \tag{9}$$

with $\mathbf{n}$ a unit vector normal to the interface and the subscript 1 or 2 denoting the different sides of the interface at which the dielectric constant and the gradient of the voltage profile are evaluated.

The induced (2D) charge density on the nanowire surface can be obtained from the following expression:

$$n(\mathbf{r}) = \frac{\epsilon(\mathbf{r})}{e}\mathbf{n} \cdot \nabla V, \tag{10}$$

with $\mathbf{n}$ the unit vector normal to the nanowire surface this time around. This induced charge density can be related to the capacitance $C(\mathbf{r})$ (locally) through the relation: $C(\mathbf{r}) = en(\mathbf{r})/V_{\mathrm{g}}$.

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
