# Peer review of "Gate-induced decoupling of surface and bulk state properties in selectively-deposited Bi2Te3 nanoribbons"

_SciPost Physics_

## Round 1 · Referee Report · Anonymous (Referee 1) · 2021-10-5

Report

The authors studied transport in micro- and nanoribbons of the Bi2Te3 topological insulator, grown by MBE following a technique developed previously in the group, the selective growth area growth approach. They focus on transconductance measurements using top gates, down to low temperature (T about 2K). They characterize their samples by measuring a microribbon in the first part of the paper and investigate the quantum transport properties and the formation of 1D subbands in a 200nm wide nanoribbon in the second part of the paper. The paper combines both experimental results and numerical simulations.

Considering the difficulty to fabricate such devices and to realize this kind of experiments, the results are of very good quality and can be considered as state-of-the-art results.

Nevertheless, this work raises major criticism concerning its novelty and the main results shown in the paper were already reported or discussed in the literature in the last years, some of these reports being cited in the present work (see for instance PRB 97, 035157; Nat. Nanotechnol. 11, 345; PRL 110, 186806). I couldn’t identify any significant step ahead beyond what has been published so far in the present version of the manuscript. In order to meet the novelty requirements, a clear (new) message should be formulated, supported by a (new?) set of data.

Apart from this, I have some important and less important comments:

1) About the existence of 1D subbands in the nanoribbon: the mobility and the density are known so that it is possible to extract the mean free path l_0 and to compare it with the perimeter of the nanoribbon. Thanks to the relation µ=el_0/(hbar k_F), I could roughly estimate l_0 to be about 20 nm for the microribbon. This value, compared to the perimeter of the nanoribbon, precludes the formation of any quantized 1D subbands as presented in the paper. In other term, the strong scattering induces a strong coupling between each subbands and the transverse wave vectors cannot be considered as good quantum number anymore. The description of the band structure in terms of 1D subbands fails down quickly as soon as l_0 becomes smaller than the perimeter (see for instance PRB 97, 075401).

2) About the topological (or trivial) nature of the phase shift: there is a confusion about the topological nature of the phase shift in the gate voltage dependence of the AB oscillations. It should be clearly stated that such a phase shift has a trivial origin and can be attributed to confinement effects only as shown in PRB 97, 075401. Similar measurement in (topologically trivial) carbon nanotubes should also lead to such phase shifts.

3) The nature of the 2DEG at the origin of the AB oscillations: the author claim that “Our analysis shows evidence of quantized transverse-momentum states on the perimeter of the TI nanoribbon and the electrostatic model treatment allows to distinguish these features from bulk effects or conventional two dimensional space charge layers without spin-momentum locking.” If the influence of the bulk is shortly (maybe not enough for the nanoribbon) discussed, an extended discussion about the nature 2DEG is lacking in the manuscript, where a simple mention of the spin degeneracy g_s of the 2DEG is made. A more detailed discussion should highlight this important result. Moreover, the fitting scenario corresponding to g_s=2 and a capacitance twice as large as the one considered here should be discussed and excluded to support the conclusion of the paper. As measured in the microribbon, such a larger capacitance would roughly correspond to the capacitance of the top surface and would support the scenario of a conventional 2DEG.

4) Quantum interferences: there is again here some confusion. According to the value of the phase coherence length, we should be between D=1 and D=2. At two dimensions, one expects tau_phi propto T-1 leading to L_phi propto T-1/2 providing that we are in the diffusive regime (L_phi propto tau_phi1/2) that precludes the formation of 1D subbands. If we are at D=1, we should have tau_phi propto T-2/3 leading to L_phi propto T-1/3 in the diffusive regime and L_phi propto T-2/3 in the ballistic regime. A clear discussion of the different cases is lacking in the paper.

5) Universal Conductance fluctuations: in the analysis of the gate voltage dependence of the AB oscillations, there is no mention of the influence of universal conductance fluctuations of the surface states and of the bulk states. In the diffusive regime, such quantum interference should nevertheless lead to fluctuations that are, in amplitude, comparable to AB oscillations.

Less important points:

- A clear definition of C(s) is lacking in the paper.
- According to their simulations, the authors show that in the case of the wide ribbon, the total capacitance is not the average capacitance integrated along the perimeter but rather the top capacitance for the nanoribbon so that the model of the effective capacitance fails to reproduce the measured conductance. It is stated that the main reason for that is the screening by bulk charge carriers. For the nanoribbon nevertheless, the model of the effective capacitance works well. This is surprising since the bulk carrier density should not vary significantly between a micro- and a nanoribbon. Does the aspect ratio (100nm dielectric for a 200nm wide ribbon vs 15nm thick dielectric and 1µm wide ribbon) causes such a large difference between the micro- and the nanoribbon? This point should be discussed in the manuscript.
- The bulk doping is about 7,6 – 4,5 10^21 cm-3. This are some very large values for such a thin structure. We could naively expect that in such thin nanostructures, the bulk would be close to the depletion regime. What is the screening length?
- What about the reproducibility? The fabrication and growth technique used by the authors should allow the fabrication of measurement of different devices. Do the authors have some measurements on other devices?
  • validity: -
  • significance: -
  • originality: -
  • clarity: -
  • formatting: -
  • grammar: -

Author:  Daniel Rosenbach  on 2021-11-07  [id 1919]

(in reply to Report 1 on 2021-10-05)
Category:
answer to question

Dear reviewer, thank you for your feedback on our manuscript. The points you mention were helpful to have another critical view on our manuscript and we would like to address your questions in this response. Also you will find a new and revised version of our manuscript, where we have elaborated on your valuable remarks. We will refer to these changes to our manuscript within our answers.

The Referee writes: The authors studied transport in micro- and nanoribbons of the Bi2Te3 topological insulator, grown by MBE following a technique developed previously in the group, the selective growth area growth approach. They focus on transconductance measurements using top gates, down to low temperature (T about 2K). They characterize their samples by measuring a microribbon in the first part of the paper and investigate the quantum transport properties and the formation of 1D subbands in a 200nm wide nanoribbon in the second part of the paper. The paper combines both experimental results and numerical simulations.

Considering the difficulty to fabricate such devices and to realize this kind of experiments, the results are of very good quality and can be considered as state-of-the-art results.

Nevertheless, this work raises major criticism concerning its novelty and the main results shown in the paper were already reported or discussed in the literature in the last years, some of these reports being cited in the present work (see for instance PRB 97, 035157; Nat. Nanotechnol. 11, 345; PRL 110, 186806). I couldn’t identify any significant step ahead beyond what has been published so far in the present version of the manuscript. In order to meet the novelty requirements, a clear (new) message should be formulated, supported by a (new?) set of data.

Our response: Our aim is to develop a scalable approach to define topological insulator-based quantum devices combining CMOS-compatible processes and state-of-the-art nanotechnology. To achieve scalability without the need of harmful top-down processing we established a selective-area growth platform and use molecular beam epitaxy for the deposition of TI layers. With this approach, we have been able to investigate devices of different cross section and dimensionality with a very good reproducibility. For related research, see Adv.El.Mat. 2020, 6, 2000205, Nanotechnol. 31 325001, Nat. Nanotechnol. 14, 825–831, Sci. Adv. 7, 26 (abf1854), arXiv:2012.15118.

In addition to applying a scalable approach, we would like to stress the novelty in our analysis on a consistent set of micro- and nanoribbon devices where we could successfully disentangle the bulk and surface response using electrostatic gating. The model we use to describe the field-effect in both type of devices includes considerations from previous publications (mentioned in both your comment and our manuscript) but generalized for systems with inhomogeneous Fermi level pinning and non-negligible bulk contributions in the overall charge depletion or accumulation. Furthermore, our analysis takes into account the differences due to the phase-coherence length being larger (in the case of the nanoribbon) or smaller (in the case of the microribbon) than the perimeter of the ribbon (more on that in our comments below). Together with the scalability of devices we are convinced our analysis proves that our approach is useful for future research on 3D TI quantum devices.

In more detail we would like to point out the differences in direct comparison to the publications mentioned in your comment:

PRB 97, 035157 (Ziegler et al.): In this publication no major bulk contributions are considered and the Fermi energy (of HgTe) can be tuned into the bulk band gap. The model used to determine the effective capacitance is therefore unsuitable for our setup and requires modifications. In contrast, the determination of the effective capacitance we use includes a non-homogeneous Fermi pinning on the perimeter of the nanoribbon as well as bulk contributions.

Nat. Nanotechnol. 11, 345 (Jauregui et al.): Nanowires investigated here are grown using a vapor-solid method and nanowires are individually transferred and contacted. Our selective-area growth approach using MBE might result in nanoribbon devices with seemingly inferior properties (bulk charge density) but will ultimately be the more scalable approach for quantum technologies.

PRL 110, 186806 (Dufouleur et al.): Also in this publication the authors grow nanowires from the vapor phase. The scalability of our approach is to be highlighted in this regard again. Furthermore, while containing an analysis of AB oscillations, this publication does not consider electrostatic gating.

The Referee writes: Apart from this, I have some important and less important comments: 1) About the existence of 1D subbands in the nanoribbon: the mobility and the density are known so that it is possible to extract the mean free path $l_0$ and to compare it with the perimeter of the nanoribbon. Thanks to the relation $\mu=e l_0 /(\hbar k_F)$ , I could roughly estimate $l_0$ (elastic mean free path) to be about 20 nm for the microribbon. This value, compared to the perimeter of the nanoribbon, precludes the formation of any quantized 1D subbands as presented in the paper. In other term, the strong scattering induces a strong coupling between each subbands and the transverse wave vectors cannot be considered as good quantum number anymore. The description of the band structure in terms of 1D subbands fails down quickly as soon as $l_0$ becomes smaller than the perimeter (see for instance PRB 97, 075401).

Our response: The estimate of $l_0$ is based on the mobility value and charge carrier density from the microribbon data. For the evaluation of the possible evolution of quantized 1D subbands these values should not be used. We showed earlier (Adv.El.Mat. 2020, 6, 2000205) that the charge carrier densities and mobility values for the topologically protected surface states can differ significantly in these bulk doped wires. We can unfortunately not access $l_0$ for the topologically protected surface states of our nanoribbons directly due to the bulk background, but assume that it is increased compared to the bulk value. Furthermore, it is more important to consider the phase-coherence length rather than the elastic scattering length when analysing AB oscillations. The phase-coherence length we address within our manuscript (Fig. 4 a) and b)).

As outlined in Dufouleur et al., PRB 97, 0754 (2018), reducing the elastic mean free path results in a considerable damping of the Aharonov-Bohm oscillations. However, a closer inspection of Fig. 6 in that article, which shows the conductance variations for zero and half flux quantum, also reveals that even for large disorder strength the Aharonov-Bohm amplitude does not vanish completely. Referring to our measurements shown in Fig.~3, one finds that the amplitude of the Aharonov-Bohm oscillations are smaller by three orders of magnitude compared to the total conductance. Thus, the signal is indeed small due to many elastic scattering events, i.e. diffusive transport. In fact, in the first experiments on Aharonov-Bohm rings made from Au (Webb,et al., Phys. Rev. Lett., 54, 2696-2699 (1985)), clear oscillations were observed, even when the transport was in the diffusive regime. However, increasing the temperature inelastic scattering events are more frequent, thus the phase-coherence decreases and eventually leads to a complete suppression of the Aharonov-Bohm oscillations, as confirmed by our experiments. Our setup differs in a sense from the planar ring structures, since for the three-dimensional TI ribbon the diffusive transport is driven along the wire. However, this does not naturally imply that the elastic mean free path should be shorter than the perimeter of the wire. But, even if that would be the case, the Aharonov-Bohm effect can still be expected, as the reference provided by the referee show.

The Referee writes: 2) About the topological (or trivial) nature of the phase shift: there is a confusion about the topological nature of the phase shift in the gate voltage dependence of the AB oscillations. It should be clearly stated that such a phase shift has a trivial origin and can be attributed to confinement effects only as shown in PRB 97, 075401. Similar measurement in (topologically trivial) carbon nanotubes should also lead to such phase shifts.

Our response: Thank you for pointing this out, the phase shift indeed is in general a signature of a surface-state subband shifting above or below the Fermi level in a flux-periodic manner. This can be the case for topologically trivial systems as well. We will make sure to properly mention this fact in an updated version of our manuscript. Nevertheless, our further analysis shows that this signature fits perfectly with the behaviour of a spin-nondegenerate massless surface state spectrum of a TI ribbon (this point we will discuss further in our response to your next comment). In these system (unlike in topologically trivial systems), we can reach a topologically nontrivial phase in this way (see Cook and Franz, PRB, 84, 201105 (2011), for example). So, indirectly, this phase shift is a signature of a topological phase (transition).

The Referee writes: 3) The nature of the 2DEG at the origin of the AB oscillations: the author claim that “Our analysis shows evidence of quantized transverse-momentum states on the perimeter of the TI nanoribbon and the electrostatic model treatment allows to distinguish these features from bulk effects or conventional two-dimensional space charge layers without spin-momentum locking.” If the influence of the bulk is shortly (maybe not enough for the nanoribbon) discussed, an extended discussion about the nature 2DEG is lacking in the manuscript, where a simple mention of the spin degeneracy $g_s$ of the 2DEG is made. A more detailed discussion should highlight this important result. Moreover, the fitting scenario corresponding to $g_s=2$ and a capacitance twice as large as the one considered here should be discussed and excluded to support the conclusion of the paper. As measured in the microribbon, such a larger capacitance would roughly correspond to the capacitance of the top surface and would support the scenario of a conventional 2DEG.

Our response: Using the Poissons solver we made use of the assumption that there is no electric field (no relative gate potential) inside the ribbon structure (the metallic surfaces effectively screen the electric field, see comment number 2, less important points). The only difference in between the micro- and nanoribbon devices therefore is that there either is or is no coherent state on the perimeter of the ribbon. Our estimation of the phase-coherence length, based on the temperature dependence of the Aharonov-Bohm conductance fluctuations, indicates that, at around 2K, the perimeter of the microribbon is much larger than the phase-coherence length, while the perimeter of the nanoribbon is comparable.

The very fact that we observe AB oscillations in the nanoribbon sample shows that there are coherent surface states on the whole perimeter of the nanoribbon device. With this in mind, the consideration of an effective capacitance on the whole perimeter of the nanoribbon is justified and only with $g_s = 1$ does this consideration result in a very good fit that is consistent with our understanding of the system. We admit that this point could be discussed better in the manuscript, explicitly discussing and excluding the possibility of a 2DEG with $g_s=2$ being responsible for the AB oscillations we find. That we did in our updated version.

The Referee writes: 4) Quantum interferences: there is again here some confusion. According to the value of the phase coherence length, we should be between D=1 and D=2. At two dimensions, one expects tau phi propto T-1 leading to L phi propto T-1/2 providing that we are in the diffusive regime (L phi propto tau phi1/2) that precludes the formation of 1D subbands. If we are at D=1, we should have tau phi propto T-2/3 leading to L phi propto T-1/3 in the diffusive regime and L phi propto T-2/3 in the ballistic regime. A clear discussion of the different cases is lacking in the paper.

Our response: It is indeed an interesting point in which regime the phase-coherent transport takes place in our structures. In Fig. 4 (a) in our manuscript the damping of the Aharonov-Bohm amplitude with temperature is shown. As depicted in Fig. 4 (b), the temperature dependence of the amplitude can be fitted very well by an $\exp(-a T^{1/2})$ dependency. This implies that the phase-coherent transport takes place in a two-dimensional regime. In fact, this dependency is quite plausible. The exponential decrease shown in Fig. 4 b) is fitted to the experimental values at temperatures from about 3 K on. From that fit we deduced a phase-coherence length of 360 nm at 1 K. Thus, for temperatures larger than 3 K the phase-coherence length is definitely smaller than the perimeter of the nanoribbon. This implies that the phase coherent transport takes place in the two-dimensional regime if one includes the propagation along the ribbon. However, we would like to stress that a phase-coherence length smaller than the perimeter does not imply that phase coherence around the perimeter is completely suppressed and no Aharonov-Bohm type oscillations can be observed.

The Referee writes: 5) Universal Conductance fluctuations: in the analysis of the gate voltage dependence of the AB oscillations, there is no mention of the influence of universal conductance fluctuations of the surface states and of the bulk states. In the diffusive regime, such quantum interference should nevertheless lead to fluctuations that are, in amplitude, comparable to AB oscillations.

Our response: In previous publications we could show that the universal conductance fluctuations are much smaller in magnitude (O~e-5 G0) than the AB oscillations (O~e-3 G0) (Adv.El.Mat. 2020, 6, 2000205). Furthermore, angle-dependent magnetoconductance measurements ruled out that oscillations in the parallel field orientation might be UCFs rather than AB.

The Referee writes: Less important points: - A clear definition of C(s) is lacking in the paper.

Our response: In section 2.2 we write: "The geometry of devices investigated induces a non-homogeneous electric field distribution and the capacitance on the nanoribbon perimeter P becomes a position dependent value C(s), wheres= [0,P]." Or are you rather asking how C(s) is determined from the V/Vg (y,z) profile we simulated using a Poissons solver? We added an explicit relation between $s$ and $(y,z)$ to make the meaning of $C(s)$ precise, as defined in the appendix.

The Referee writes: - According to their simulations, the authors show that in the case of the wide ribbon, the total capacitance is not the average capacitance integrated along the perimeter but rather the top capacitance for the nanoribbon so that the model of the effective capacitance fails to reproduce the measured conductance. It is stated that the main reason for that is the screening by bulk charge carriers. For the nanoribbon nevertheless, the model of the effective capacitance works well. This is surprising since the bulk carrier density should not vary significantly between a micro- and a nanoribbon. Does the aspect ratio (100nm dielectric for a 200nm wide ribbon vs 15nm thick dielectric and 1µm wide ribbon) causes such a large difference between the micro- and the nanoribbon? This point should be discussed in the manuscript.

Our response: You are right, the surface charges on the top surface screen the charges on the bottom surface. This is indeed formulated not very well in the current version of our manuscript (page 8, line 24). The comparison in between the effective capacitance and the experimentally determined capacitance within the last paragraph of section 'Gate-dependent microribbon Hall measurements' is in fact misleading and will be removed in an updated version. As mentioned in the paragraph before, it is the phase-coherence that allows us to use an effective model for the nanoribbon device. For the microribbon device we consider a parallel plate capacitor model applied to the top surface (including surface and bulk) due to the screening of the electric field by the (top!) surface charges.

The Referee writes: - The bulk doping is about 7,6 – 4,5 $10^{21}$ cm-3. This are some very large values for such a thin structure. We could naively expect that in such thin nanostructures, the bulk would be close to the depletion regime. What is the screening length?

Our response: We calculated the bulk doping concentration in the microribbon based on the two-dimensional bulk charge carrier densities of $7.6 \times 10^{13} \mathrm{cm}^{-3}$ and a thicknesses of 10 nm ($10^{-8}$ cm) and obtained a value of $7.6 \times 10^{21} \mathrm{cm}^{-3}$. The value is lower than the value given in the report, i.e. the carrier concentration is not that exceptionally large and corresponds to typical values given in literature. Furthermore, in topological insulators the topologically protected surface states are confined to a depths of typically one nanometer. This implies that for our layer thickness the top and bottom layers basically don't overlap and that it is well feasible that bulk charge carriers are present.

The screening length for a three-dimensional system can be calculated from:

$$ k_0^2= \frac{e^2}{\epsilon_0} \frac{m^*}{\hbar^2 \pi^2} (3 \pi^2 n_{3D})^{1/3} $$
with $m^*=0.58 m_0$. The screening length is then given by $1/k_0=0.11$\,nm.

The Referee writes: - What about the reproducibility? The fabrication and growth technique used by the authors should allow the fabrication of measurement of different devices. Do the authors have some measurements on other devices?

Our response: On the same chip, we have fabricated micro- and nanoribbon devices as presented in our manuscript. Next to these devices we have as well been investigating nanoribbon ‘kinks’. These nanostructures merge two nanoribbons under a specified angle. Using selective area growth this will not lead to any disruption of the film at the interface (see e.g. tri-junctions in arXiv:2012.15118). These structures also show clear AB patterns. The analysis of these AB oscillations as a function of the gate (only the kink region has been gated) and an in-plane magnetic field, however, are not that easily discussed as the magnetic field can only be oriented perpendicular to the effective cross section of one of the two ‘arms’(nanoribbons) of the kink structure at the same time. A full understanding of this set of measurements is still pending.

Also for reproducibility, please refer to the references given in our response to your general comment.

---

## Round 1 · Referee Report · Anonymous (Referee 2) · 2021-10-20

Report

The authors report on the electrical investigation of gated Bi2Te3 ribbons. Whereas I do not have an issue with the quality of presented measurements, the main issue is to determine in which respect the presented data and conclusions surpass the state-of-the-art in the field. From the presented manuscript it is not clear which groundbreaking experimental discovery it details or that it opens a new pathway in an existing or a new research direction. The manuscript may be considered for SciPost Physics Core if the authors can meet its acceptance expectations, which, unfortunately, at present the manuscript does not.
  • validity: -
  • significance: -
  • originality: -
  • clarity: -
  • formatting: -
  • grammar: -

Author:  Daniel Rosenbach  on 2021-11-07  [id 1920]

(in reply to Report 2 on 2021-10-20)

Dear reviewer, thank you very much for your honest feedback.

While topological surface states have now been realized in many different material systems (the number of topological materials increases rapidly) and different experiments, one of the main difficulties to define a state-of-the-art is to find a material system that is scalable for more advanced experiments. We would like to argue that we add to the understanding of one of the most used and well known topological insulators, Bi2T3. We worked on an electrostatic model that treats systems with inhomogeneous Fermi level pinning and bulk conduction, which helped us to identify quantized transverse momentum states in molecular beam epitaxy (MBE) grown Bi2Te3 nanowires and provide evidence for their topological nature. Furthermore, we provide results for a size dependent transition based on phase-coherence by studying devices of different (lateral) dimensions.

Besides, using a selective area growth approach via MBE our results open up the pathway to scale up from single devices to networks of nanoribbons that can both be controlled using a top gate and an external magnetic field, while all the processes involved for the sample fabrication are compatible with standard CMOS processing. That our processes are indeed scalable and provide good reproducibility you can see following previous pubications: Adv.El.Mat. 2020, 6, 2000205, Nanotechnol. 31 325001, Nat. Nanotechnol. 14, 825–831, Sci. Adv. 7, 26 (abf1854), arXiv:2012.15118.

Following above mentioned arguments we are of the opinion that our report adds sufficient novelty to be considered for publication in SciPost. In our manuscript we have identified inconsistencies based on comments from reviewer number 2 and have adapted to a more concise, updated manuscript, where we highlight the novelty of our approach.

---

## Editorial Decision

resubmitted